# Animal Models for Studying Congenital Transmission of Hepatitis E Virus

**DOI:** 10.3390/microorganisms11030618

**Published:** 2023-02-28

**Authors:** Kush Kumar Yadav, Scott P. Kenney

**Affiliations:** Center for Food Animal Health, Department of Veterinary Preventive Medicine, The Ohio State University, Wooster, OH 43210, USA

**Keywords:** hepatitis E, pregnancy models, rabbit, mouse, pig, chicken, immune correlates

## Abstract

One of the most intriguing issues in the hepatitis E virus (HEV) field is the significant increase in mortality rates of the mother and fetus when infection occurs in the second and third trimesters of gestation. A virus that is normally self-limiting and has a mortality rate of less than one percent in otherwise healthy individuals steeply rises by up to 30% in these pregnant populations. Answering this pivotal question has not been a simple task. HEV, in general, has been a difficult pathogen to understand in the laboratory setting. A historical lack of ability to efficiently propagate the virus in tissue culture models has led to many molecular aspects of the viral lifecycle being understudied. Although great strides have been made in recent years to adapt viruses to cell culture, this field remains behind other viruses that are much easier to replicate efficiently in vitro. Some of the greatest discoveries regarding HEV have come from using animal models for which naturally occurring strains of HEV have been identified, including pigs and chickens, but key limitations have made animal models imperfect for studying all aspects of human HEV infections. In addition to the difficulties working with HEV, pregnancy is a very complicated biological process with an elaborate interplay between many different host systems, including hormones, cardiovascular, kidneys, respiratory, gastrointestinal, epithelial, liver, metabolic, immune, and others. Significant differences between the timing and interplay of these systems are notable between species, and making direct comparisons between animals and humans can be difficult at times. No simple answer exists as to how HEV enhances mortality in pregnant populations. One of the best approaches to studying HEV in pregnancy is likely a combinatorial approach that uses the best combination of emerging in vitro and in vivo systems while accounting for the deficiencies that are present in each model. This review describes many of the current HEV animal model systems and the strengths and weaknesses of each as they apply to HEV pregnancy-associated mortality. We consider factors that are critical to analyzing HEV infection within the host and how, despite no perfect animal model for human pregnancy mortality existing, recent developments in HEV models, both in vitro and in vivo, are advancing our overall understanding of HEV in the pregnant host.

## 1. Introduction to Human Pregnancy

### 1.1. Three Trimesters of Pregnancy

While scores of textbooks are completely devoted to the topic of human reproductive physiology, we hope to convey enough basic pregnancy physiology to present the reader with sufficient knowledge to vet key factors that may contribute to the hepatitis E virus (HEV) pathogenesis and to gain an understanding of key differences in animal model systems of pregnancy. Human pregnancy is a complex biological process that includes the interplay of several host organ systems resulting in the development of a fertilized egg into a fully formed baby, which is delivered via live birth over the course of approximately 266 days. Human pregnancy is divided into three trimesters, each lasting approximately 90 days. Through the first two–four weeks of the first trimester, nutrition and waste are processed via diffusion through the endometrial lining. The mother’s liver is responsible for the removal of waste chemicals from the fetus, such as bilirubin. As the trimester continues, the external layer of the embryo merges with the endometrium, and the placenta is formed. The placenta is the first organ to develop and is the largest fetal organ to play a vital role in the health of both the fetus and its mother, with the placenta becoming a physical barrier, which viruses such as HEV must cross to infect the developing fetus. The nutrient and waste needs of the embryo and fetus are managed by this organ, while nutrients are passed to the placenta via the mother’s blood, and waste is removed from the same channel. Not all species’ placentas are created equally as humans and primates possess a hemochorial placental type that is similar to rodents and rabbits, while horses, pigs, and ruminants have an epitheliochorial placenta, and carnivores have an endotheliochorial placenta [1]. The primary difference among placental types is in the number of cell layers in the interhaemal area and the ease with which the transfer of macromolecules between the maternal and fetal blood occurs, making it an important factor with respect to the difference in placental permeability between animal species. Interestingly, limb buds, eyes, the heart, and the liver are formed in five weeks but are typically not functional. The body is formed by the end of eight weeks when the fetus is about five centimeters in length. Through the second trimester, the fetus grows to nearly 30 cm. All organ systems continue to grow. The placenta takes over the roles of nutrition, waste excretion, and the production of estrogen and progesterone, as the corpus luteum has deteriorated at this point. From this point throughout the delivery, the placenta continues functioning as the major organ. During the third trimester, the fetus increases from 3 to 4 kg and is around 50 cm long. This trimester produces the most rapid growth during pregnancy. Although multiple organ systems continue to grow until birth, some systems, such as the nervous system and liver, keep developing even after birth. Readers seeking more detail on reproductive physiology are referred to [2,3].

### 1.2. Immunological Changes during Pregnancy to Understand HEV Related Effects

The role of hormonal and immunological changes during pregnancy has long been implicated as potentially exacerbating HEV pathogenesis [4,5]. Pregnancy immunology has three stages that are characterized by distinctive processes [6].

Initially, the embryo implantation and early stages of placentation lead to a strong inflammatory response throughout the first and the beginning of the second trimester. This phase is similar to wound healing, primarily due to embryo implantation, invasion, and the vascularization of trophoblast cells into the maternal endometrium [7,8]. Inflammation is necessary to ensure that sufficient restoration of the uterine epithelium occurs with the removal of cellular debris and tissue remodeling. Therefore, a pro-inflammatory stage can be seen during the first trimester of pregnancy, in which Th1-type cytokines are in the majority [8,9].

The second immunological stage of pregnancy is marked by the speedy growth of the fetus. The prevalence of Th2-type cytokines establishes an anti-inflammatory state so that maternal–fetal interface harmony can be maintained [10]. The maternal immune system engages in a balancing act during gestation, retaining tolerance to the fetus while maintaining innate and adaptive immune mechanisms against microbes [9,11,12]. Trophoblasts which form the outer placental layer, form an immunological cloaking device around the fetus as they do not express major histocompatibility complex (MHC) proteins [13], making the placenta impervious to T-cell mediated injury while expressing unique human leukocyte antigen G (HLA-G) molecules that actively suppress natural killer (NK) cells [14].

In the last stage, which occurs very late in the third trimester, fetal development finishes, and the baby’s birth is attained through a restored state of inflammation that is backed by Th1-type cytokines and immune cell invasion into the myometrium [15]. This inflammatory process supports the tightening of the uterus, which leads to the expulsion of the fetus and placenta. Hence, pregnancy shifts between pro-inflammatory and anti-inflammatory states [16]. The involvement of specific cellular and molecular elements characterizes each immunological stage. The balance between pro-inflammatory and anti-inflammatory responses is controlled by several regulatory mechanisms at the maternal–fetal interface that can significantly alter the outcomes of viral infection.

In summary, alterations in the Th1:Th2 cell ratio throughout pregnancy can be seen with a significant tilt in the direction of Th2 cells. Particularly during the initial 20 weeks of pregnancy, which is considered an important phase for fetus survival, the levels of most cytokines are suppressed. How these initial immunological modulations translate into increased mortality risks of patients infected with HEV during the second and third trimesters is not yet clear-cut with the existing data.

### 1.3. Hormonal Changes during Pregnancy to Understand HEV Related Effects

Hormonal factors throughout pregnancy have been suggested to play a significant role in HEV mortality [4]. Pregnancy or placental hormones can be grouped into two main categories: steroid hormones and protein hormones. Steroid hormones include progestins, which bind to the progesterone receptor, and estrogens derived from either fetal androgens, placental progestins, or other steroid precursors. The placentae of all mammals are thought to secrete progestins, with the quantity secreted varying widely by species. In humans, the placenta can produce enough progestin to maintain pregnancy in the absence of ovaries, while in species such as pigs, pregnancy would fail if the ovaries were removed. In humans, placental estrogen is in the form of estriol. Within the placental protein hormone category, there are chorionic gonadotropins, placental lactogens, and relaxin. Placental chorionic gonadotropins are only recognized in primates and equid species. Placental lactogens are molecular relatives of prolactin and the growth hormone. They have been identified in primates, ruminants, rodents, and lagomorphs but not in other species. Relaxin is produced in the corpus luteum, the placenta, and the uterus in females. The sites of relaxin production vary by species: some animals only secrete relaxin from a single source, while others utilize all three sites [17]. As the roles of hormones on HEV replication are not well studied, it is quite possible that some of the lesser-appreciated pregnancy hormones might significantly contribute to HEV pregnancy mortality. Hormones, including progesterone, estrogen, and human chorionic gonadotropin (HCG), rise throughout pregnancy to potentially modify immune regulation and viral replication [18,19,20,21]. Cell-mediated immunity suppression can be seen because of the hormones [22]. HCG has been shown to regulate cell-mediated immunity [23], estrogen causes the contraction of the thymus and reduces the populations of CD4 and CD8 [24], and progesterone blocks T-cell development and inhibits Th1 cell development while promoting the development of the Th2 cell [25,26,27]. When considering the roles of hormones on HEV and pregnancy, the researcher should use caution to fully understand the endocrinology of the animal model and how it may differ from human pregnancy hormones.

## 2. HEV Background

In the 1980s, non-A, non-B hepatitis (NANB) was identified to be HEV [28]. HEV was found to be 7.2 kb in size and a quasi-enveloped, single-stranded RNA virus with a positive-sense genome. The absence of peplomers at the membrane-associated HEV virions confirmed that the virus was quasi-enveloped, similar to the hepatitis A virus [29]. Three open reading frames (ORF 1, 2, and 3), edged by 5′ and 3′ untranslated regions (UTR), have been defined in the viral genome [30]. A 7-methylguanylate (m^7^G) cap is present at the end of the 5′ UTR, which drives cap-dependent genome translation. The 3′ UTR possesses a poly-adenylated tail [30]. Of the three conserved ORFs, ORF1 encodes for the non-structural polyprotein, which comprises functional domains that are essential for viral genome replication [31,32]. Bicistronic subgenomic RNA is responsible for the synthesis of overlapped ORF2 and ORF3 [33]. The viral capsid protein is encoded by ORF2 and possesses three domains: S (shell), M (middle), and P (protruding). Recently, ORF2 was shown to encode a secreted form (ORF2^S^), for which the biological role needed to be further elucidated [34,35,36]. ORF3 encodes a phosphoprotein that appears to play an important part during viral particle morphogenesis [37]. A fourth open reading frame has been reported in gt1 human and rodent HEV strains. Both ORF4s overlap ORF1 but contain little sequence conservation of the proteins between viruses. In human HEV, ORF4 appears to enhance replication during endoplasmic reticulum stress and, when ectopically expressed, enhances gt3 replication in cells, and the function of rodent HEV ORF4 is unknown [38,39,40].

### 2.1. HEV New Classification

HEV has recently undergone classification updates and continues to evolve with the discovery of new HEV strains. The recent reclassification of the family *Hepeviridae* comprises two subfamilies: Orthohepevirinae and Parahepevirinae. The former infects multiple terrestrial and arboreal animals that consist of four genera. *Paslahepevirus balayani* comprises eight genotypes (gt1–gt8), four of which currently cause the most cases of hepatitis E in humans (gt1, gt2, gt3, and gt4) [41]. HEV gt1 and HEV gt2 are obligate to humans, while HEV gt3 and HEV gt4 have zoonotic importance, and thus, the transmission occurs primarily from swine to humans via pig and undercooked pork products [42] but can also be transmitted by other species such as rabbits and deer. The three other genera, *Avihepevirus*, *Rocahepevirus*, and *Chirohepevirus*, largely circulate in birds, rodents, and bats, respectively.

### 2.2. Geographical Distribution of HEV

Although HEV is described as an emerging infectious agent, it is the established main cause of acute viral hepatitis (AVH) globally [43]. More than 20 million infections are anticipated to occur globally each year, resulting in nearly 70,000 infections leading to death [44]. Developing countries such as Asia, Africa, and Latin America are the primary regions that report major epidemic outbreaks resulting in deaths related to exposure to faecally-contaminated water that leads to outbreaks and sporadic cases of hepatitis E [44,45]. In addition, hepatitis E in these geographical regions is attributable to the human-associated HEV gt1 (Asia, Africa, and Latin America) and gt2 (sub-Saharan Africa and Mexico). However, the worldwide spread of zoonotic strains (gt3 and gt4) has lately been recognized as a reason for sporadic hepatitis in medically susceptible patients and the general population, including in high-income countries [46,47,48,49,50]. Several findings imply that hepatitis E is a zoonotic disease that is associated with pigs, rabbits, rats, and camels, which act as reservoirs for human infection [51,52,53].

### 2.3. Pregnancy Mortality Reported Due to HEV

Higher case-fatality ratios due to acute liver failure (ALF) in pregnant women in their third trimester remain a typical pathognomonic feature of hepatitis E outbreaks [54,55,56,57]. Alongside the fecal-oral route of transmission, parenteral and vertical routes of transmission have been described for HEV [58,59,60]. Limited data are available on the mother-to-child spread of HEV infection from India. It is reported that HEV infection produces serious liver disease with higher mortality rates in fetuses and neonates [61]. The survivors of vertically-transmitted HEV infection demonstrate self-limiting clinical progression with short-lasting viremia [61].

Evidence from 18th-century Europe demonstrates epidemic jaundice with an excess of illness and death among pregnant women and their infants, potentially representing the first historically documented accounts of HEV [55,62]. It was in the mid-1950s when the first retrospective study serologically validated a hepatitis E outbreak in Delhi, India; however, molecular proof hints that HEV may have already been flowing through humans for hundreds of years [30,56,63,64,65]. During the Delhi epidemic, the hospital-based study documented a 10% maternal case-fatality rate alongside miscarriage, stillbirth, or neonatal death in 56% of the infants of women with HEV infection. Notably, expiring and surviving infants were reported as having jaundice [66].

It has been over six decades since HEV was recognized as a reason for infectious hepatitis, and several subsequent studies have similarly reported high rates of maternal, fetal, and neonatal illness and death in pregnancies affected by HEV; however, the mechanism for pregnancy mortality has not yet been fully identified [67]. A report from epidemiological disease models has suggested that HEV might be responsible for almost 2400–3000 stillbirths per year in developing countries, with additional fetal deaths associated with antenatal maternal mortality [44,68,69]. Hepatitis E-related preterm delivery in mothers is common, with worse neonatal survival [68,70]. The reports from a 2002 outbreak in the Central African Republic indicated premature deliveries by all of the serologically confirmed hepatitis E (*n* = 7) pregnant women. Briefly, three stillborn cases, with one of them macerated and a fourth baby, were reported dead within a few minutes of delivery [71]. Similarly, in a 1993–1994 outbreak in Islamabad, Pakistan, all the newborns from acute hepatitis E mothers comprised 50% (4/8) of the fatalities [54]. During a 2008–2009 hepatitis E outbreak in Tongi, Bangladesh, pregnant women diagnosed with jaundice were more than two times as likely as non-jaundiced pregnant women to miscarry or deliver a stillborn baby [72]. In New Delhi and Chennai, India, two separate hospital-based prospective studies reported that 15–50% of the live-born infants of mothers with hepatitis E died within the first 1-week post-partum [68,73]. Likewise, the 2010–2011 outbreak in Sudan reported 14 intrauterine deaths and nine premature deliveries among 39 pregnant hepatitis E cases [74].

Similarly, in one report of eight infants born to HEV-infected mothers in their third trimester, seven infants were born at term, and one was born prematurely at 34 weeks gestation. Anti-HEV IgG was detected in all the infants, and HEV RNA was detected in five. One baby was jaundiced at birth with elevated serum ALT levels, and four were anicteric with elevated ALT levels. After birth, two babies developed hypothermia and hypoglycemia and died within 24 h. On autopsy and liver biopsy, one of these infants had massive necrosis [59]. Another report on nineteen infants born to HEV-infected mothers resulted in vertically transmitted HEV infection being seen in fifteen at birth, IgM anti-HEV positivity being detected in twelve, and HEV RNA being detected in ten, with three experiencing short-lasting IgG anti-HEV positivity because of maternal antibody spread across the placenta. Additionally, seven HEV-infected patients were reported with icteric hepatitis, five with anicteric hepatitis, and three with high serum bilirubin with normal liver enzymes. Seven infants died in the first week after birth, including one death due to prematurity. Of the nine survivors with follow-up data, five were HEV RNA positive. Interestingly, HEV RNA was not detectable in three patients by 4 weeks of life, in one by 8, and in the remaining infants by 32 weeks. The self-limited disease was seen in all survivors. They reported that HEV transmission from the mother to the fetus caused high neonatal mortality [61]. Therefore, the situation necessitates effective animal models for the study of HEV infection.

To date, there have been no animal models that could recapitulate all the clinical manifestations seen in humans (Figure 1). Rabbits, pigs, non-human primates, chickens, mice, rats, ferrets, and Mongolian gerbils have been used for pathogenesis, vertical transmission, and vaccine efficacy studies. However, none of them have been able to fulfill the characteristics that require them to be the ideal model for HEV pregnancy mortality.

## 3. Characteristics of an Ideal Model Recapitulating HEV Infection in Humans

An appropriate model system for the study of HEV during pregnancy must address several important criteria, which we summarize here. (a) The model should be susceptible to genotypes 1–4 *Paslahepevirus* strains of HEV as these genotypes are most consistently detected in current human cases. (b) The model should recapitulate HEV pathology, including liver lesions and similar sites of replication. To fully recapitulate the human condition, researchers should strive to reproduce the classical pathological symptoms (pregnancy pathology) associated with the disease. Simply injecting a virus into the placenta of a pregnant animal has a high likelihood of causing abortion and other morbidities but likely does not mimic a true systemic infection. (c) A reverse genetic model system should exist for the virus strains. The ability to assess the contributions of viral genetic factors to mortality is vital to understanding the underlying mechanisms of disease and for the ability to produce a consistent starting inoculum that can normalize results across studies. (d) The animal model should mimic humans with a chorional placenta, and the pregnancy hormones should be similar. Ideally, researchers should strive to recapitulate the human pregnancy environment as closely as possible. Many animals vary greatly in their hormonal cycles and do not come close to matching the human process. (e) Models should contain a similar gestation period or one that can be converted to trimesters in terms of development. Obviously, an animal model that lasts for 266 days might not be the most time-efficient method of study; however, the protracted length of HEV infection to symptom onset suggests that models with short time spans might miss critical points in the lengthy HEV replication cycle. Animal models with very short gestation cycles would likely need significant optimization to find times that are critical for causing HEV mortality. (f) The ability to measure innate and adaptive immune responses to the HEV virus and to manipulate these responses under experimental infection conditions is needed. The host immune system performs a substantial role in both pregnancy and managing pathogen infection. The ability to observe these events in real-time HEV infection in the pregnant host and experimentally manipulate the host response is critical to understanding pregnancy mortality. (g) This method is also cost-effective and widely available for research use. As with all research model systems, the ability to perform enough replicates per study and the ability to confirm results in multiple labs is good scientific practice. As with most animal model systems, no current HEV model fulfills all the listed requirements, but a few existing models work well for some of the requirements.

To address the adverse consequences of maternal hepatitis E on the fetus and to determine if these are either the results of maternal health complications or fetal HEV infection, several animal models and a few in-vitro models to study the pathogenesis of HEV to reproduce vertical transmission of HEV in humans have been explained with detailed experimental findings.

### 3.1. Rabbit Model

Rabbits have a hemodichorial and bidiscoid type of placenta [1,75]. The gestation period of a rabbit is 29–35 days. Pregnancy maintenance is achieved by the corpus luteum in rabbits and requires estrogen from ovarian follicles and prolactin [76]. The first rabbit HEV (rHEV) strain was isolated in 2009 from a Chinese farm and classified as a gt3 *Paslahepevirus* strain [77]. Since then, numerous rabbit strains have been identified globally, indicating rabbits to be a natural host for HEV [78,79,80,81]. In rabbits, acute HEV infection can be seen with fecal viral shedding, elevated liver enzymes, histopathological changes in the liver, HEV-specific antibodies, and viremia [82]. Likewise, the experimental inoculation of rabbits with the CHN-BJ-RB14 HEV strain, which demonstrates chronic HEV infection, is defined by the shedding of the virus in feces and the presence of the virus in blood [83]. Moreover, chronic inflammatory cell infiltration and portal fibrosis were demonstrated in liver histopathology [83]. Experimentally infected rabbits with rHEV also exhibited extrahepatic replication in diverse tissues, such as the brain, heart, lungs, stomach, intestine, kidney, and placenta. Similarly, the extrahepatic replication of HEV in the kidneys of chronically infected rabbits from the above study presented lesions in the organ, which was thought to be induced by the replication of the virus [84]. Furthermore, the replication of HEV in the ovaries of rabbits after the intraperitoneal injection of swine gt4 HEV has recently been proposed [85].

In 2010, forty-two specific pathogen-free (SPF) rabbits were divided into 11 groups, with 1 group acting as the negative control in the first-ever pathogenesis study. The intravenous (IV) inoculation of these SPF rabbits with several strains of rHEV (GenBank No. FJ906895, FJ906888, FJ906896, FJ906893) was performed. Between 10^1^ and 10^7^ genome equivalents (GE) of inocula were utilized. Rabbits were actively infected after HEV inoculation with fecal shedding from 1 to 2 weeks post-inoculation (wpi), and viremia followed at 4 wpi. In addition, at 14 wpi (the end of the study), some of the rabbits demonstrated fecal and serum HEV RNA. During this late period of fecal shedding, elevated alanine aminotransferase (ALT) levels were also observed. The enzyme levels peaked from 9 to 11 wpi with a four-fold elevation from the control level. For pathological signs of HEV infection, liver histology was conducted. Multifocal lymphocytic infiltrations and local hepatocellular necrosis were detected. All rabbits injected with non-passaged rHEV strains were seroconverted 3 months post-inoculation. Hence, the susceptibility of rabbits to rHEV strains was confirmed, and the disease severity in the rabbit was concluded to be dose-dependent [86].

In 2012, Cheng et al. reproduced the findings of the Ma et al. study and demonstrated that the rabbits, which injected IV with rHEV strains (GenBank No: JQ065065, JQ065068) had comparable clinical manifestations of acute hepatitis E. Furthermore, the authors also administered the rHEV strains in 15 rabbits orally, in addition to IV inoculation. The infectivity of the orally administered virus was minimal, with only two rabbits displaying fecal shedding and seroconversion [87]. The experiments above demonstrate that the pathogenesis of rHEV in rabbits is similar to acute HEV infection observed in humans, with fecal shedding of HEV RNA, viremia, seroconversion, evident histopathological changes, and raised ALT levels, although an IV administration with a high viral dose is required. Hence, the above-mentioned data imply that the rabbit is currently one of the closest models for recapitulating acute HEV infection in humans.

Interestingly, around 2014, Han et al. demonstrated that rabbits injected with the rHEV isolate CHN-BJ-RB14 had a 9-month fecal shedding of HEV RNA. Chronic inflammatory cell infiltrations and obvious portal fibrosis seen during histopathology were the indicators of chronic HEV infection in rabbits. The lengthy viremia and fecal shedding in rHEV-infected rabbits resembled human chronic HEV infection. Seroconversion at approximately 5 wpi was seen in all rabbits and higher antibody levels were maintained by all the rabbits, until the end of the study, with an exception of one rabbit who seroconverted at 22 to 25 wpi and became undetectable thereafter. A similar study has been reported in humans as well, which is thought to be related to host immune status [88].

In 2015, Xia et al. demonstrated that the rabbit HEV strain CHN-BJ-RB14 was able to produce high mortality with the vertical transmission in newborn kits when used to experimentally infect pregnant rabbits. To investigate whether HEV infection could produce any pregnancy defects, HEV-infected (group1) and non-infected (group 2) female rabbits were permitted to copulate with healthy male rabbits. Their pregnancy status was determined by both a rise in serum progesterone and the palpation of embryos performed by professional breeders. They demonstrated that all HEV-infected rabbits (6/6) of group 1 remained non-pregnant, but 9/12 rabbits of group 2 became pregnant. Their results indicate that the rate of infertility was considerably higher in the HEV-infected rabbits (100%, 6/6) than in non-infected rabbits (25%, 3/12). When confirming HEV infection in group 1 rabbits through the detection of HEV RNA in the feces, they showed persistent or intermittent fecal shedding of HEV RNA from 3 days post-inoculation (dpi) in the six rabbits. The duration of HEV infection, as characterized by fecal virus excretion varied from 6 weeks post-inoculation (wpi) to 15 wpi (experiment ends). Although six rabbits were infected with HEV, viremia was observed only in two rabbits, and increases in ALT and aspartate aminotransferase (AST) levels were roughly two-fold higher than the baseline level that was detected in only one rabbit. Seroconversion to anti-HEV antibodies occurred at 3–7 wpi except for one rabbit that demonstrated no seroconversion.

Further studies that demonstrate how the rabbit model is capable of linking the adverse effects of HEV during pregnancy include the study where six pregnant rabbits from group 2 were infected with HEV, while three pregnant uninfected rabbits were used as a control group. They found that two/six pregnant rabbits had a miscarriage while three of the remaining four HEV-infected pregnant rabbits died. In contrast, the control groups experienced no miscarriage (0/3) or death (0/3). Liver tissue histopathology in pregnant rabbits with HEV infection was also conducted and demonstrated bridging necrosis or piecemeal necrosis and the infiltration of inflammatory cells via hematoxylin and eosin (HE) staining. In lung sections, they observed pathological manifestations of pulmonary edema, and in kidney sections, focal lymphocytic infiltration surrounding the blood vessels appeared. In contrast to them, the control groups did not appear to have any gross histopathology lesions in similar tissues compared to HEV-infected pregnant rabbits. Positive and negative strands of HEV RNA in the liver tissues from HEV-infected pregnant rabbits suggested active virus replication. HEV RNA was not detected in the lung, kidney, and heart tissues of any of the rabbits, either infected or not infected. The six pregnant rabbits infected with HEV showed persistent fecal virus excretion from as soon as 3 dpi until their death or until the experiment ended. An increase in ALT and AST was seen in three out of four of the remaining rabbits without miscarriage. However, there was no seroconversion to anti-HEV antibodies in the two rabbits. In addition, they mentioned that the control groups were negative for anti-HEV antibodies and fecal HEV RNA and had normal ALT and AST levels.

Positive and negative strands of HEV RNA were discovered in the placental tissues of HEV-infected pregnant rabbits. They also demonstrated positive staining for the HEV antigen in placental tissue through immunohistochemistry (IHC). This finding provided confirmatory evidence of HEV replication in the placental tissue. Control group placental tissue sections were negative for the HEV RNA and HEV antigen. To investigate the HEV vertical transmission, newborn kits from four out of six HEV-infected pregnant rabbits (two had miscarriages) were studied. Fecal HEV RNA was detected from the primary defecation of all newborn rabbits born from infected mothers. They recommended that the antibodies against HEV that were identified in these newborns were probably induced by infection rather than passively acquired from their HEV-infected mothers because the seroconversion to anti-HEV antibodies occurred at 3 months of age in those newborns. Hence, they confirmed that HEV infection in newborn kits from infected mothers supported the vertical transmission of HEV. In contrast, newborns from pregnant rabbits with no HEV infection were negative for both fecal HEV RNA and anti-HEV antibodies. This reported the first-ever recapitulation of the typical pregnancy outcomes seen in human HEV infection in an animal model, demonstrating a higher mortality rate, miscarriage, and vertical transmission.

In 2017, Wang et al. demonstrated that rabbits infected with gt3 rHEV produced both chronicity and kidney injury. Six SPF Japanese white rabbits (R1–R6) were injected IV with 10^6^ (R1, R2), 10^5^ (R3, R4), and 10^4^ copies (R5, R6) of the CHN-BJ-RB14 strain of rHEV gt3. They detected viremia/fecal shedding of HEV at 1 wpi. Of the six rabbits (R1–R6), R1 and R2 demonstrated persistent HEV infection with fecal virus shedding for 40 and 20 weeks, respectively, but R1 did not seroconvert to anti-HEV until its death at 40 wpi. Furthermore, positive and negative-stranded HEV RNA was identified in the kidneys, suggesting virus replication in this tissue. Sections of the liver confirmed the infiltration of inflammatory cells in the portal area, along with venous dilation and fibrosis, suggesting chronic hepatitis. Renal tubule cavities experienced injury due to the protein casts and serious infiltration of lymphocytes; plasma cells in the renal interstitium indicated clear kidney injury. Positive staining was observed when IHC was performed to detect HEV ORF3 proteins in the R1 kidney. Hence, this positive staining confirmed that lesions were produced by HEV replication in the kidneys.

Hence, Wang et al. conducted a study with the inocula of 10^4^, 10^5^, and 10^6^ copies of CHN-BJ-RB14 which were, respectively, introduced IV into two naïve SPF rabbits per dose. Rabbits injected with 10^6^ copies of rHEV consequently developed persistent HEV infection. An exception took place in one rabbit which presented at 40 weeks of infection with signs of portal fibrosis and chronic inflammatory cell infiltration [84]. In summary, the studies by Wang et al., and Han et al. suggested the dose-dependent development of chronic HEV infection in rabbits and demonstrated that rabbits might be the appropriate animal model for chronic HEV infection. In addition, many specifics are still to be studied that are crucial to the chronicity of HEV infection, and the factors to be considered, especially those linked with the host’s immune status.

In addition, extra-hepatic replication has been mentioned in rHEV-infected rabbits. Briefly, the positive/negative HEV RNA and HEV ORF2 antigen were identified in the rabbit’s brain, heart, lung, stomach, intestine, kidney, and placenta [82,83,89]. Recently, in 2019, Tian et al. utilized rabbits to examine the mechanism of virus invasion into the nervous system. The brain and spinal cord demonstrated the presence of the HEV RNA and HEV ORF2 protein. Pathological changes included the perivascular cuffs of lymphocytes and microglial nodules, which were also associated with CNS infections. Thus, these results identify rabbits as a credible model for studying neurological disorders associated with HEV [90]. Hence, multiple studies have defined the rabbit as the closest animal that recapitulates clinical manifestations as seen in humans. Pregnancy mortality, the vertical transmission of HEV, chronic HEV infection, and extra-hepatic manifestations have all been studied and are still ongoing in rabbits.

In 2017, An et al. evaluated the replication of strain HB-L3, gt4 swine HEV (90.9% homology to a Beijing human strain) in the ovary and explored structural and molecular alterations stimulated by the intraperitoneal injection of HEV in rabbits. At 28 dpi, one of four ovary samples in the HEV-injected group was positive for positive-stranded HEV RNA. However, no ovary sample showed negative strands of HEV RNA. At 49 dpi, two of four ovary samples in the HEV-infected groups demonstrated both positive-stranded and negative-stranded HEV RNA. HEV RNA-positive ovaries (signal for both ORF2 and ORF3) were identified in the HEV-infected group at 28 and 49 dpi. Interestingly, the ovarium ovum demonstrated a positive signal as well [85]. At 28 dpi, HEV RNA-positive ovary epithelial cells revealed scattered necrosis and dropped off. In addition, they reported that scattered necrosis was observed in follicular cells of primordial follicles. Furthermore, they demonstrated that programmed cell death in follicle cells and oocytes was supported by HEV infection. They suggested the ovary as one of the extrahepatic replication sites of HEV. This claim was further supported by the relation of germ cell apoptosis with the existence of HEV RNA and antigens in ovarian tissue. Notably, they suggested the vertical transmission of HEV with a newly proposed mechanism that demonstrated HEV infection and replication in the different stages of the ovum. They implied that HEV antigens present in the ovum could progress into the fertilized ovum after insemination and then into the embryo [85].

In 2019, Li et al. studied the pregnancy results that could pertain to different HEV genotypes and if prevention could be achieved by the HEV 239 (Hecolin, 26 kDa recombinant polypeptide expressed by the Escherichia coli system derived from the 368–606 amino acid segment of the HEV gt1 ORF2) vaccine. Forty-two female rabbits were split into seven groups, with two groups administered with a preventative vaccine and one group with PBS. Except for the negative control, all other groups were either inoculated with rabbit HEV gt3 (CHN-BJ-R14), swine HEV gt4 (CHN-SD-SW2), or human HEV gt3 (CHN-SH-W). Interestingly, pre-exposure to the HEV 239 vaccine before copulation resulted in no HEV infections. However, all other rabbits that were inoculated with different strains from rabbits, swine, and humans successfully resulted in adverse pregnancy outcomes. Furthermore, newborn rabbits that were born to vaccinated individuals were free of HEV in comparison to the survived individuals from HEV-infected females that demonstrated HEV infection [91].

Rabbits are small animal models that can be used for cross-species infection, pathogenesis, pregnancy as well as vaccine studies. Several studies are ongoing to establish an animal model mimicking the HEV clinical manifestations seen in humans, to understand the pregnancy-related pathology, and to identify the novel characteristics of HEV replication. Although convincing results from previous studies have proved rabbits to be one of the best models to utilize for HEV pregnancy mortality at the current time, there are specific drawbacks to the model. Studies in which rabbits were experimentally inoculated with human strains of gt3 HEV confirmed seroconversion but were unsuccessful in demonstrating replication and fecal shedding [87,92]. In addition, rabbits were also resistant to human gt1 and gt2 HEV infection [87], which limits our ability to understand the gt1 specific characteristics and the pathology-related to it. Currently, we are unable to discern if HEV pregnancy mortality in rabbits is unique to the rabbit strain of the virus or if human (gt1–gt4) strains would cause similar pregnancy mortality if they could replicate efficiently in rabbits.

In summary, the rabbit model is currently the best HEV model for studying pregnancy mortality. Rabbits were infected with a naturally occurring gt3 strain of HEV that also infects humans. Rabbits have a systemic infection resulting in liver lesions as well as the ability of the virus to progress to chronicity and to cause pregnancy mortality (Figure 2). Rabbits are commercially available, and researchers can study the lapine immune response in depth. The rabbit is not vulnerable to infection by either gt1 or gt2 strains.

### 3.2. Non-Human Primates (NHPs) Model

Cynomolgus monkeys have a haemomonochorial and bidiscoid type of placenta that is very similar to the human placenta [93]. Gestation times vary by species ranging from 133 days in owl monkeys to 164 days in rhesus macaques and up to 240 days in chimpanzees. Hormonal regulation in NHPs is most similar to humans within the discussed animal models. Progesterone and estrogens are the principal steroid hormones produced by the placenta [94]. The experimental infection of NHPs demonstrates susceptibility to gt1, gt2, gt3, and gt4, although they are not the natural host for HEV [95,96,97,98]. The first NHP used for experimental HEV infection studies was cynomolgus macaques. HEV was identified in 1983 when pooled stool samples derived from Afghan patients were ingested by a human volunteer while studying a hepatitis outbreak. The volunteer showed symptoms of AVH, and virus-like particles (VLPs) were seen in his stool samples. Cynomolgus macaques were inoculated IV with stool samples from the volunteer. Interestingly, macaques reported histopathological and enzymatic hepatitis, which was attributed to the causative agent. In addition, cynomolgus macaques excreted VLPs in their feces, and VLP-specific antibody responses were also observed.

After the successful experimental infection in cynomolgus macaques, other NHPs were evaluated for the study of HEV. Chimpanzees, cynomolgus, and rhesus macaques were used to compare HEV infection. Surprisingly, HEV gt1 demonstrated higher susceptibility to chimpanzees in comparison to rhesus and cynomolgus macaques. In contrast, the virulence of the same strain was comparatively higher, as suggested by elevated liver enzymes [96].

Furthermore, HEV infects different NHPs, including pig-tailed macaques, vervets, owl monkeys, squirrel monkeys, and patas monkeys but tamarins the infections are dubious because infection was not demonstrated in all the inoculated animals [99,100]. Most of these animals were IV inoculated because oral inoculation had failed to develop an infection in most study trials. Moreover, the research suggested 10,000-fold higher oral dose requirements than IV to develop an active HEV infection in cynomolgus macaques. [98]. Overall, the NHPs recapitulate the infection seen in humans, but the existing difference between species marks the difference in fecal viral shedding, liver enzyme regulations, and microscopic lesions in the liver.

To examine the virulence between several HEV genotypes, rhesus monkeys were used. They found that gt3 HEV was significantly less virulent than human gt1 and gt2 [96]. To study HEV pathogenesis during pregnancy, Tsarev et al. in 1995 demonstrated that experimental HEV infection in pregnant rhesus monkeys resulted in no fulminant hepatitis as seen in pregnant women infected with HEV. Further, he demonstrated no transmission of HEV to the young and showed proof of naturally developed antibodies to HEV. In addition, pregnant rhesus macaques (*n* = 6) in the first, second, or third trimester of pregnancy and non-pregnant rhesus macaques (*n* = 4) were injected IV with approximately 10 (5.5) ID_50_ of HEV to experimentally reproduce the typical fulminant hepatitis of pregnant women seen during HEV infection. The comparison of biochemical, histopathological, and serological profiles compared between pregnant and non-pregnant rhesus monkeys did not show a rise in the severity of hepatitis in the pregnant animals. In addition, they showed a normal range for hematology and serum clinical chemistry values in all animals throughout the study. There was no proof of newborn infection with HEV in the offspring; thus, the vertical transmission of HEV in rhesus monkeys was not seen. Before inoculation, the animals were screened for anti-HEV antibodies by a standard ELISA, and two rhesus monkeys (1 pregnant, 1 non-pregnant) had naturally occurring anti-HEV antibodies, which were confirmed by a competition ELISA with hyperimmune chimpanzee serum [101]. Hence, these animals exhibited an anamnestic response when confronted with HEV.

To demonstrate the vertical transmission, three pregnant rhesus macaques were infected with HEV gt4 (KM01 strain derived from HEV-infected swine stool; 5.1 × 10^5^ copies/mL [102]. Fecal virus shedding and viremia demonstrated active HEV infection. One of the pregnant rhesus macaques infected with HEV had premature delivery resulting in fetal death. Interestingly, the fetal kidney, liver, spleen, and intestines were positive for HEV RNA. They suggested how the cumulative effect of a compromised innate immune system, lowered progesterone levels and turns in immune status may provoke HEV infection and end in harmful pregnancy effects [102].

To study chronic HEV infection and to mimic immunocompromised patients, cynomolgus macaques were treated with tacrolimus, a potent calcineurin inhibitor immunosuppressant, and were determinedly inoculated with gt3 HEV strain [103]. Mild enhancement in the liver enzymes, persistent RNA viremia, viral fecal shedding, and severe hepatic lesions were directed toward chronic HEV infection [103].

NHPs share the most similar course of pregnancy to humans in terms of placenta composition, pregnancy hormones, and length of gestation. Although multiple studies have been conducted using NHPs, the disadvantage of not being the natural host of HEV, along with limited clinical presentations of the disease, makes it an unsuitable model for studying human HEV infection (Figure 3). Furthermore, NHPs are the most expensive model for research purposes; as such, it is also limited in research usage due to related ethical concerns.

### 3.3. Mouse Model

The chorioallantoic placenta of the mouse is discoid in shape and hemotrichorial in type [93,104,105]. The gestation period of mice ranges from 20–27 days. In rodents, pregnancy is maintained by the corpus luteum, which continuously produces steroid hormones in comparison to other species, such as humans and ruminants, where the main source is the placenta [106,107]. As with most non-human primates, mouse progesterone peaks in late pregnancy and subsides prior to the birthing process, whereas, in humans, progesterone continually increases through parturition. Many human pathogens do not infect mice naturally because of divergent host adaptation during evolution. To study specific human pathogens in mice, genetically engineered mice were used to express human factors that make them vulnerable to the pathogen [108,109,110]. Recently, some of these studies have shown robust HEV infection by the experimental injection of human liver chimeric mice with HEV gt1 and gt3 [110,111,112].

In 2017, when Upa-SCID (*n* = 3) mice were injected with cell culture-originated HEV gt3 Kernow C1-P6 (1.7 × 10^6^ IU/mouse), viral RNA was observed in the feces of injected mice (2/3), with approximate titers of 6.2 × 10^4^ IU/mL [110]. In addition, when HEV gt1 Sar55-infected chimpanzee’s stool suspension was intrasplenically injected in mice (*n* = 2), both demonstrated HEV RNA (stool and plasma) after1 wpi. When the magnitude of the infection was compared, gt1 was much more robust than gt3 HEV [110]. Supportively, the progressive infection has been observed with uPA^+/+^Nod-SCID-IL2Ry^−/−^ mice when injected with pooled fecal and liver-derived inoculum (HEV gt3) [111].

In 2017, Sayed et al. used humanized FRG mice to evaluate several ways of inoculation. Fecal viral RNA shedding and late viremia were observed in mice when intrasplenic inoculation (purified patient fecal suspension, HEV gt3) was administered. Similarly, plasma from the constant patient developed into disease (1/2). Notably, the amount of virus in the feces was much less when a direct comparison was made to stool inoculation mice. Moreover, when the human strain (gt1, sar55) was used in mice, RNA levels were reported to be 10^4^ IU/mL from week 2 post-injection (p.i). However, humanized uPA-SCID and FRG mice failed to demonstrate the infection when the virus was orally inoculated [113].

In 2019, Yang et al. reported effective HEV infection in pregnant BALB/c mice. They proposed a pregnant mouse model to study the clinical consequences of HEV infection. Briefly, mice were injected IV with HEV gt4 (strain KM01) (1 × 10^4^) at 3 days (early pregnancy, *n* = 9), 10.5 days (middle pregnancy, *n* = 9), and 14 days (late pregnancy, *n* = 9) post-inoculation mimicking the trimesters of human pregnancy. Negative control group mice (*n* = 9) were inoculated with PBS after copulation. The fecal shedding of HEV viral RNA was seen at 3 to 5 dpi; however, viremia was depicted in pregnant mice from 5 to 7 dpi. The screening of tissues for HEV viral RNA (positive and negative-stranded) resulted in an HEV RNA-positive liver, spleen, kidney, and colon. As expected, negative control group samples were all negative for HEV viral RNA. Interestingly, delivery was normal in all the mice during early and late pregnancy. Miscarriages (7/8, 87.5%) were noted in mice injected with HEV at the halfway point of gestation, with one exception of normal delivery. For further identification of the HEV replication in mice, HEV antigens were detected by IHC, which were found to be disseminated in the liver, spleen, kidney, and colon, in both conditions, either delivered normally or in case of abortion. As expected, the negative control mice demonstrated HEV negative. Hence, their findings firmly suggest that HEV can replicate in pregnant BALB/c mice [114].

Even though no maternal deaths were reported in the study, seven mice injected with HEV in mid-pregnancy developed miscarriages. Notably, there was a smaller number of fetuses in the HEV =-infected group in comparison to the negative control group [early (35 fetuses), middle (3 fetuses), or late (61 fetuses)]. They demonstrated that the imbalance of the Th1/Th2 immune condition in HEV-infected mice could be related to the elevated frequency of miscarriage. Fine modulation of Th1/Th2 stability is a crucial cause for the safeguarding of fetuses as opposed to abortions in mice [115]. They reported a higher Th1/Th2 ratio (9.48 fold) in HEV-infected miscarriage mice than in uninfected mice with normal delivery [114].

They also reported the vertical spread of HEV from the mother to the fetus. During early, middle, and late pregnancy, HEV RNA (both positive and negative strands) could be seen in HEV-injected mothers. In addition, the uterus and placenta of all HEV-infected mice (usual deliveries and terminated pregnancies) showed HEV antigens. Inflammatory exudates and hemorrhages were apparent upon histopathological examination, suggesting that severe inflammatory responses were seen in the uterus of terminated mice (middle pregnancy). To investigate if the fetus experienced HEV infection, fetal livers from mothers diseased with HEV at early (fetus = 35), middle (fetus = 3) or late (fetus = 61) pregnancy periods were evaluated for HEV viral RNA (positive and negative strands) infection. Apart from two, all the tested fetuses were positive, suggesting that the BALB/C mouse model is adequate for pursuing the vertical spread of HEV from the mother to the fetus but requires independent verification [114].

Furthermore, in 2021, Yang et al. demonstrated miscarriage, non-pregnancy, and uterine damage in HEV gt4 (KM01)-infected BALB/c mice after copulation with healthy males. They suggested that uterine damage in relation to endometrial thinning enhanced inflammatory response, and increased programmed cell death could be causative factors of pregnancy pathology. Moreover, they demonstrated the recovery of uterine layers and improvement in fertility after the clearance of HEV from female BALB/c mice [116].

Although the above-mentioned data demonstrates BALB/c mice as suitable animal models for HEV vertical transmission studies, the lack of maternal mortality in these mice makes them slightly deviate from the clinical manifestation seen in humans (Figure 4). Data from the uPA and FRG mice are convincing and suggest they could be a better small animal HEV model. However, the deficiency of the adaptive immune system in these animals makes them inappropriate for immune-related mechanistic studies against HEV [117,118]. More work with the BALB/C model is necessary to evaluate its ability to sustain gt1, gt2, and gt3 infections and what contribution they play to pregnancy mortality in this model.

While mice have been a workhorse model system for demonstrating many complex biological functions, more study is needed to determine if a truly representative HEV pregnancy model can be established in these animals. The significant genetic manipulations often needed to achieve HEV growth in the mouse make them a very artificial model. Their fast and prodigious growth and short reproductive cycles make them ideal for performing large numbers of experiments quickly. Due to their extensive usage as model animals, there are many commercially available reagents for mapping almost every aspect of a murine immune response, and mice are amenable to techniques such as immune cell transfer experiments. The further establishment and optimization of HEV mouse models are warranted due to the benefits mentioned above.

### 3.4. Pig Model

Pigs have an epitheliochorial and diffuse type of placenta [1]. Swine gestation lasts for 114 days. The pig placenta mainly produces estrogen hormones. For successful pregnancy establishment, estrogen secretion from the placenta between days 10 and 15 of pregnancy is very essential in pigs [119]. In 1997, swine HEV was identified in the United States, and it was recognized as the initial strain of HEV found in animals. Since then, it has been detected globally in domestic and wild swine [120]. Briefly, swine HEV was primarily noticed via the recognition of anti-HEV seropositive adult pigs, followed by prospective research on piglets from an Illinois herd that resulted in the rescue of a completely new virus [120]. The successful spread of the novel virus to SPF pigs and recovering similar viruses from the experimentally infected SPF pigs satisfied Koch’s postulates [121]. Swine is the most studied animal to date for the expansion of an animal model for studying HEV infection.

In 2001, Williams et al. demonstrated extrahepatic replication sites of HEV in a swine model. Two pigs from group 1 (18 pigs, IV inoculations of swine HEV), group 2 (19 pigs, IV inoculations of human HEV, US-2 strain), and group 3 (17 pigs, uninoculated controls) were necropsied at 3,7,14, 20, 27, and 55 dpi. They confirmed the occurrence of positive-strand HEV RNA in many tissues by RT-PCR. From 3 to 14 dpi, positive-stranded viral RNA was detected in the tissues of swine HEV-infected pigs. From 3 to 20 dpi, tissues were found positive for human HEV RNA in inoculated pigs. However, they claimed that the positive viral strand RNA recognition may not have been from the replicating virus since serum also tested positive for viral RNA. Interestingly, even after the disappearance of viremia, they were able to detect positive-strand viral RNA between 20 and 27 dpi in swine HEV-inoculated pigs and at 27 dpi in human HEV-inoculated pigs. Thus, they confirmed that the tissues and organs that were positive for viral RNA (positive strand) in the absence of viral RNA in the serum were an indicator of success for viral RNA replication. Furthermore, tissues that were demonstrated as positive for viral RNA (positive stranded) were also tested for negative strand viral RNA using negative strand-specific RT-PCR, which gave positive results. Liver, lymph node, small intestine, and colon tissues demonstrated a longer duration of HEV replication when compared to any other tissues. Notably, the major extrahepatic replication sites of HEV were determined to be lymph nodes and the intestinal tract [122].

In 2001, Halbur et al. performed a comparative pathogenesis study of infected pigs with HEV recovered from a pig and a human from the above groups. They revealed that two pigs from each group had no indication of liver enzyme elevation or clinical disease. However, they demonstrated enlarged hepatic and mesenteric lymph nodes in both HEV-injected groups. In addition, multifocal lymphoplasmacytic hepatitis was observed in 15 of 18, 16 of 19, and 9 of 17 pigs from groups 1 to 3, respectively. Focal hepatocellular necrosis was observed in 10 of 18, 13 of 19, and 5 of 17 pigs from groups 1 to 3, respectively. At 20 dpi, human HEV-inoculated pigs demonstrated hepatic inflammation and hepatocellular necrosis, which peaked in severity, and at 55 dpi, they were found to be moderately severe. In contrast, at 55 dpi, swine-HEV-injected pigs demonstrated the absence of or nearly resolved hepatitis lesions. Furthermore, they demonstrated that all HEV-injected pigs seroconverted to anti-HEV IgG. Additionally, it was interesting to know that pigs developed comparatively more severe and persistent lesions when injected with the US-2 strain of human HEV than swine HEV based on microscopic evaluations. Thus, they proposed that xenograft recipients would be on the probable risk list from HEV-infected pig livers or cells [123].

In 2003, pregnant gilts, their fetuses, and their offspring were studied to dissect the effect of swine HEV. Swine HEV was IV inoculated to 12 gilts at day 79 post-breeding, and six gilts were utilized as a negative control. Between five and six gilts (four injected and one or two controls) were euthanized on three separate days as follows: 91 d of gestation 12 dpi, or from 17 to 19 d after farrowing (55 dpi; 2 controls and 4 inoculated). Four 8–10-day-old piglets from each of the six sows that farrowed were necropsied at 46 dpi. Surprisingly, inoculated gilts/piglets showed no HEV-related clinical signs/fever. Furthermore, liver chemistry summary estimates were not distinct among inoculated and control gilts. In addition, fetal weight, length, offspring birth weight, and weight gain were studied but resulted in non-significant differences between the inoculated and control gilts. Similarly, no reproductive parameters differed between inoculated and control gilts. Furthermore, they did not see a considerable difference in the gross lesions either in gilts, fetuses, or piglets. In addition, they could not see any notable microscopic lesions in any of the piglets or fetuses [124].

At 12 dpi, the necropsy of inoculated gilts (*n* = 4) was conducted to preclude ELISA thereafter. Anti-HEV antibodies were seen in the remaining 8 gilts at 26 dpi, and all four gilts that farrowed maintained seropositivity until 47 dpi. In contrast, all control gilts were negative during the study. Their findings demonstrated that the IV inoculation of pregnant gilts with US swine HEV had no adverse effects. Vertical transmission was not seen from the dams to their fetuses, even though fecal viral shedding was noticed in the gilts. Anti-HEV IgG antibodies were seen in pigs born from swine HEV-infected dams passively from colostrum and persisted until they were 2 months old. Furthermore, fulminant hepatitis was not recapitulated in IV-injected pregnant swine. Hence, they observed subclinical hepatitis in pregnant gilts with HEV infection; however, reproductive clinical signs or lesions in the fetus or offspring could not be seen [124].

In 2005, Huang et al. generated capped RNA transcripts of full-length cDNA clones of pig HEV which they revealed to be replication competent when transfected into human hepatoma 7 (Huh7) cells and infectious after intrahepatic injection into pigs. They generated three differing cDNA clones (pSHEV-1, pSHEV-2, and pSHEV-3) by performing site-directed mutagenesis. The transfection of human liver huh7 cells by capped RNA transcripts followed by the production of ORF2 capsid and ORF3 proteins suggested the replication competence of the infectious cDNA clone. Active swine HEV infection was seen with each of these three clones when capped RNA transcripts were injected intrahepatically. They showed similar disease patterns between clones pSHEV-2 and pSHEV-3 and wild-type swine HEV, made evident by the seroconversion, viremia, and fecal shedding of viruses. Similarly, pigs injected with RNA transcripts from clone pSHEV-1, which had three non-silent mutations in the ORF2 capsid gene, seroconverted late with delayed fecal virus shedding and had unnoticeable viremia [125].

Thus, subclinical courses of infection were reported in pigs. Under experimental conditions, as mentioned above, pigs affected with swine HEV demonstrated no clinical defects but were consistently able to reproduce microscopic liver lesions that were similar to the naturally infected pigs with measurable HEV RNA in the feces, liver tissues, and bile [123]. In general, from 2 to 4 months of age, domestic pigs were typically infected by HEV, developing a transient viremia lasting from 1 to 2 weeks and fecal virus shedding from 3 to 7 weeks [126]. Naturally infected pig data demonstrated that 86% of pigs were HEV-infected by 18 weeks of age [127]. Feacally-contaminated feed and water sources are the direct mode of transmission of HEV to naïve pigs. Maternal antibodies waned in pigs around 8 weeks of age such that piglets demonstrating seropositivity against swine HEV developed IgM anti-HEV antibodies, which peaked with fecal virus shedding. At 4 months of age, IgG anti-HEV antibodies peaked, resulting in the clearance of shedding the virus in the feces. [126,127]. In contact-infected piglets, HEV RNA was detected in feces by 7 dpi, with an infectious period of approximately 49 days [128]. Hence, gt3 and gt4 HEV infections hold a subclinical course in both naturally and experimentally infected swine without any observable clinical illness or increase in the liver enzymes [123]. Swine are quickly infected via IV injection experimentally; however, oral inoculation in swine is inefficient [46].

Overall, as a model for human HEV infections, swine proficiently yield infection with gt3 and gt4 HEV and serve as a major reservoir for foodborne zoonotic HEV spread to humans. The pig is valuable for the study of different aspects of zoonotically transmitted HEV replication, pathogenesis, and cross-species infection. The size of the placenta and length of gestation (114 days) are more similar to humans than many of the other proposed models. The major drawback of the pig model regarding pregnancy is the lack of susceptibility to human gt1 and gt2, which is associated with pregnancy mortality in humans. The experimental infection of pigs with gt3 HEV during gestation did not lead to enhanced morbidity or death. Additionally, not all clinical manifestations are present in pigs infected with gt3 and gt4, and thus, it does not mimic the gross hepatic disease or lead to pregnancy mortality, restricting its usefulness in pathogenicity experiments (Figure 5).

### 3.5. Chicken Model

The avian reproductive system is significantly different than mammals which have evolved to allow fetal development in a terrestrial environment without the need for implantation and development in the host [129]. Chicken eggs take about 21 days from the time they are laid until the chicks hatch. Once released from the animal, no exogenous hormonal stimuli are received. Avian HEV was discovered in the United States in 2001 after the separation and description of the virus from chicken bile, where the chicken had hepatitis splenomegaly syndrome (HSS) [130]. In addition, the big liver and spleen (BLS) disease virus displayed similar lesions in chickens in Australia while sharing roughly 80% nucleotide sequence identity with avian HEV [131,132]. Variants of the same virus within the avian HEV clade are known to cause two syndromes (HSS and BLS). Three genotypes (gt1–gt3) are recognized for avian HEV, and phylogenetic assessments uncovered that the virus is antigenically and genetically linked to mammalian strains sharing 60% sequence similarity with human HEV strains [131].

The single known reservoir for avian HEV is chickens, with reported fecal-oral transmission between flocks. In the US, reports suggest that 71% of chicken flocks and 30% of individuals are infected by avian HEV [133]. Clinical signs such as egg production drop were noted in 20% of hens [134]. Following infection, only a small number of birds exhibited clinical signs. Regressive ovaries, serosanguinous abdominal fluid, enlarged, hemorrhagic, and necrotic livers, and enlarged spleens are the typical postmortem lesions seen in avian HEV-infected birds [135]. The experimental injection of avian HEV via IV or intrahepatic routes in chickens demonstrate fecal virus shedding and viral RNA in serum, bile, and liver samples. Furthermore, subcapsular hemorrhages, expanded lobe, and periphlebitis plus lymphocytic phlebitis are reported as liver lesions in almost 25% of infected chickens [134].

The identification of avian HEV and its similarity to HEV strains in humans permits the usage of chickens as a homologous animal model to research HEV-related pathology and replication strategies in detail. Avian HEV is limited in its host range. Chickens are not vulnerable to human or swine HEV, and the detected symptoms of HEV-positive chickens do not match with the clinical development seen in humans.

Laying chickens have been used as a model to assess vertical HEV transmission associated with gt1 infections in pregnant women. In 2007, Guo et al. demonstrated that avian HEV experimentally infected chickens contained an infectious virus in their egg whites but lacked proof of complete vertical transmission. In 2006, Guo et al. examined twenty undamaged eggs gathered at 3 weeks p.i. while making an avian HEV infectious fecal stock with a titer of 10^4^ GE mL^−1^ by IV inoculation of SPF chickens. Avian HEV in the egg was detected by performing a modified qRT-PCR procedure regularly utilized for avian influenza virus detection [136]. Five out of the twenty eggs collected had noticeable avian HEV RNA in the egg white. Koch’s postulate was confirmed by the 98% nucleotide sequence similarity reported from the positive egg white samples when compared to the original viral stock. Thus, they showed that avian HEV could be transmitted into eggs [137].

To determine whether the avian HEV in egg whites was infectious, SPF hens (*n* = 38) and SPF roosters (*n* = 2) seronegative to avian HEV were separated into three groups. Group 1 with 18 hens was inoculated IV with a 400 μL egg white sample (10^4^ GE mL^−1^), and group 2 with 18 hens was inoculated IV with 400 μL regular egg white sample (negative for avian HEV RNA by RT-PCR). Group 3 consisted of two hens and two roosters; however, only two hens were inoculated IV with 400 μL avian HEV infectious stock with a 10^4^ GE mL^−1^. In group 1, viremia and fecal virus shedding were seen from 3 wpi. By contrast, no viremia and fecal virus shedding were observed in group 2 chickens. In addition, as expected, two hens from group 3 that received an infectious stock of avian HEV had measurable viremia and fecal virus shedding, which started at 1 wpi. Chickens after avian HEV injection developed seropositivity (anti-avian HEV IgG), which was seen at 4 and 5 wpi from group-1 chickens and at 2 wpi in both hens in-group 3. As expected, group 2 remained seronegative [137].

For the establishment of the vertical transmission model for HEV, avian HEV transmission into newborn chicks was studied. From one week prior to the virus injection to 5 weeks post-injection, embryonated eggs were collected regularly from the above-mentioned group 3 chickens. Six batches of eggs (11–13 eggs/day) were hatched, and each batch was kept independently and monitored every day for fecal virus shedding and viremia for 1 week prior to the necropsy. RT-PCR results for fecal, bile, and liver samples were negative for avian HEV infection. Hence, even though egg white was positive for avian HEV, the vertical transmission was not supported due to the absence of the virus in the samples collected from 60 hatched chicks. However, they showed that avian HEV could successfully propagate into 9-day-old embryonated chicken eggs when inoculated IV with 100 GE of avian HEV. Hatched chicks necropsied at 2–3 days of age were reported as avian HEV positive for bile and liver samples. Similarly, fecal shedding was positive for 8 days of hatching. They concluded that avian HEV could enter the egg white (typically 10 mL egg white is present in chicken eggs) with a relatively high dose; however, no virus was detected in the chicks, signifying that avian HEV in eggs could not live through the initial embryonation phase (1–9 days of embryonation) [137].

The chicken model was able to evaluate vertical transmission and the causes of infertility, as seen with drops in egg production. However, the significant differences in physiology from mammals and resistance to infection with mammalian HEV make the avian HEV model difficult for its use for pathogenesis studies and to stand as an ideal model for HEV pregnancy mortality (Figure 6).

## 4. In Vitro Models

Although not a complete animal model system, placenta cell-derived models, including organoids, can be given consideration for modeling HEV pathogenesis during pregnancy. These systems can be created from human cells and tissues and mimic in vivo phenomena such as the innate immune response to viral infection and how pregnancy hormones might contribute to infection outcomes. Care needs to be taken when choosing established cell lines to ensure that specific cells are reactive to exogenous stimuli such as hormones. In 2018, Knegendorf et al. reported HEV replication and interferon responses in human placental cells. They demonstrated differential tissue-specific host responses to HEV genotypes, further enhancing our knowledge of the pregnancy pathology mechanisms adding to fatal outcomes. They noted that HEV was capable of completing its viral life cycle in placental-derived cells (JEG-3). They demonstrated that both gt1 and gt3 HEV (approximately 5 × 10^5^ GE) replicate following transfection in JEG-3 cells. Furthermore, they identified comparable attributes in liver-derived cells, as shown by extracellular and intracellular viral capsid levels, infectivity, and biophysical properties. In addition, they found that ribavirin efficiently inhibited HEV gt1 and gt3 in placental and in human hepatoma cells. They suggest that interferon-α sensitivity was comparatively less in placental cells than in liver cells for gt1, but the same was not seen for gt3. Both HEV genotypes were able to efficiently downregulate selected interferon-stimulated genes (ISGs) in the liver and placental-derived cell line. They proposed that distinctions in placenta-specific innate immune responses could be an appropriate cause for the clinical descriptions of HEV during pregnancy. Further experiments are required to analyze the core process of this occurrence. They stated that HEV was able to complete the full viral replication cycle in the placenta and speculated that this phenomenon played a role in the vertical transmission of HEV [138].

The genotype-related pathogenesis of HEV during pregnancy is indistinguishable, and a lack of efficient models has hampered fetal transmission studies of HEV. In 2018, Gouilly et al. reported HEV gt1 replication in the decidua and placenta tissue explants with observable apoptotic cells [139]. IL-6, CCL-3, CCL-4, and CXCL-10 secretion levels were highly correlated with the viral load of HEV gt1. They unveiled a genotype-specific cytokine storm relationship with the pathogenesis at the maternal–fetal interface. Impairment in the type III secretion system was highlighted as a factor that HEV gt1 needed to maintain effective replication at the maternal–fetal interface. In general, stromal cells support vascular tissue remodeling, and fetal placenta development was identified as a target for HEV gt1 [139]. Their study provides a plausible justification for the differential consequences seen during HEV gt1 and gt3 infection. Similar results were demonstrated by Mokhtar et al., 2020 in non-decidualized primary human endometrial stromal cells (PHESCs) [140]. They demonstrated the permissiveness of PHESCs to gt1 HEV infection to be more pronounced than HEV gt3 infections with impaired type III interferon response. They stated that HEV could complete full viral replication in PHESCs [140].

The above-mentioned in vitro studies and ex vivo studies such as JEG-3, PHESCs, and studies use the decidua basalis and fetal placenta, which demonstrate the relative differences in the interferon response, tissue damage and viral replication ability of gt1 and gt3 HEV. The in vitro and ex vivo studies failed to provide a complete picture of a human pregnancy. Human pregnancy is a complex mechanism supported by immune regulation. Hormonal regulation plays a collaborative role with maternal and fetal physiology. Human reproductive physiology is much more complicated and nuanced than the simple summation of cellular or isolated tissue effects. While the tissue host tropism of HEV can be identified and innate immune responses characterized cells in a dish, these were unable to provide a complete picture of the human body’s response to HEV infection. Future perspectives would be designed to a suitable experimental model system resembling in vivo tissue architecture such as organoids, spheroids, bioprinted microtissues, and organ on-chip systems [141].

## 5. Conclusions

An ideal animal model, which reliably reproduces all aspects of pregnancy mortality and pathology, associated with human HEV infection during pregnancy, does not currently exist, and finding an amenable system reaching all ideal requirements might not be feasible. Recent work has suggested the rabbit model in which rHEV gt3 strains can induce mortality of both the doe and kit, along with recapitulating many facets of HEV pathology, makes it currently one of the best models to utilize for dissecting pregnancy mortality. The further development of small animal models, such as the BALB/c mouse model, to determine if gt1 or gt2 HEV pathology can be recapitulated is warranted, and provide hope for further small animal model development in the HEV pregnancy mortality field. Experiments that compare and contrast the tried-and-true natural HEV infection models, such as swine and non-human primates, which fail to induce enhanced pregnancy mortality, and the rabbit model in which pregnancy mortality was observed might be a key piece to understanding important differences in the virus and host responses leading to HEV pregnancy mortality.

## Figures and Tables

**Figure 1 microorganisms-11-00618-f001:**
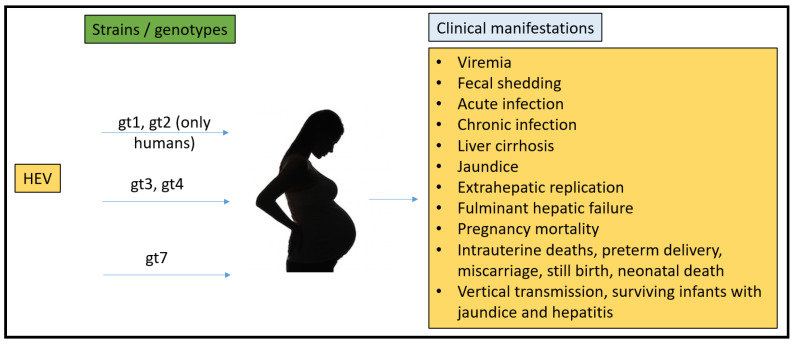
Summary of HEV strains/genotypes and overall clinical manifestations in humans.

**Figure 2 microorganisms-11-00618-f002:**
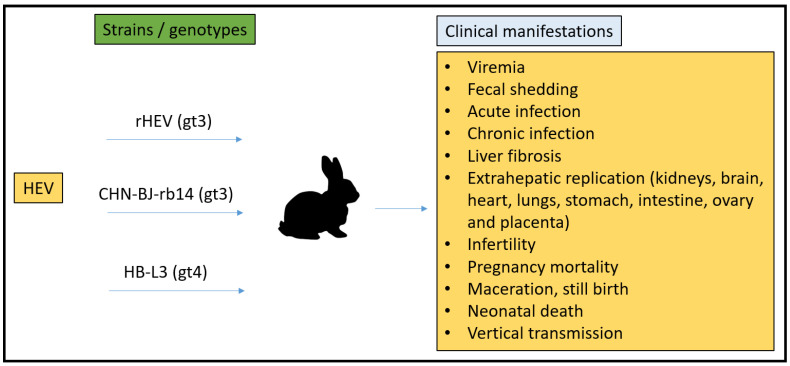
Summary of HEV strains/genotypes and overall clinical manifestations in rabbits.

**Figure 3 microorganisms-11-00618-f003:**
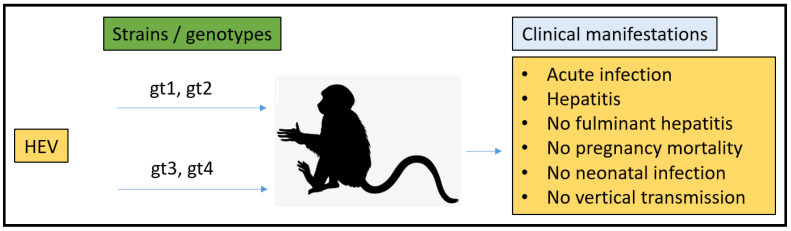
Summary of HEV strains/genotypes and overall clinical manifestations in non-human primates.

**Figure 4 microorganisms-11-00618-f004:**
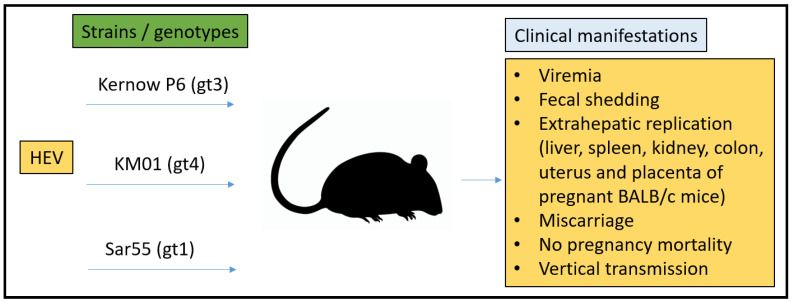
Summary of HEV strains/genotypes and overall clinical manifestations in mice.

**Figure 5 microorganisms-11-00618-f005:**
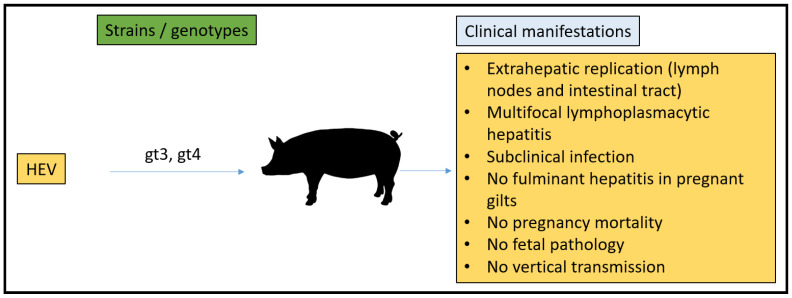
Summary of HEV strains/genotypes and overall clinical manifestations in pigs.

**Figure 6 microorganisms-11-00618-f006:**
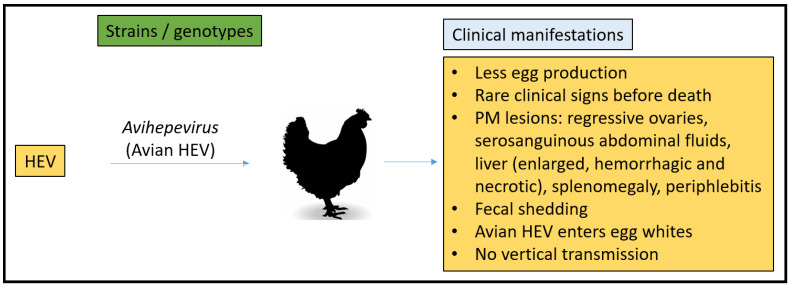
Summary of HEV strains/genotypes and overall clinical manifestations in chicken.

## Data Availability

Not applicable.

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
