# Peer review of "Animal Models for Studying Congenital Transmission of Hepatitis E Virus"

_microorganisms, 2023, doi:10.3390/microorganisms11030618_

Round 1

Reviewer 1 Report

The authors described general information related to HEV, including its structure, genetic characteristics, and then the different Animal models that have been used for studying the congenital transmission of the hepatitis E virus and the related findings.

The review provides an interesting summary of already published data, however, the following items should be addressed:

Lines 56-149 provide basic pregnancy physiology. Although the presented data are useful for general knowledge, however, they are not related to the topic of the review. This section should be either deleted or abbreviated.

In addition, a summary of the reported cases in pregnancy should be provided with the target population, geographical location, detected genotype, the tested parameters, and the outcome of the infection at the level of mother and fetus/newborn should be presented.

The tile, introduction to HEV, which includes lines 150-250 (about 100 lines) mentioned different topics related to HEV including the structure, genotypes, complications, and epidemiology in pregnant women. Therefore, this section should be divided into subtitles to make it easier for the readers to follow the presented data.

I see that the authors have reported a review about the use of animal models in studying the replication and pathogenesis of HEV in general. Unfortunately, only limited data related to the topic of pregnancy are presented. Authors should focus on the main target of the manuscript rather than describing HEV in animal models which were extensively reviewed. I know that the data related to pregnancy are not much however, spotting the light on this information is of great value to readers. For example, the title 1.4. Rabbit Model (lines 287-477). Studies of HEV replication in the ovary and transmission in pregnancy are discussed only in lines 430-477!

Similar observations are seen in the other animal models.

There are some important items that should also be included in this review such as studying the mechanisms responsible for the high morbidity of hepatitis E in pregnancy

Author Response

Microorganisms submission:

Addressing the comments from reviewers:

We would like to thank the editor for reviewing our manuscript “Animal models for studying congenital transmission of hepatitis E virus” and considering it to be applicable for the special issue “Emerging Pathogens Causing Acute Hepatitis”.

We would like to thank the reviewers for their feedback that helped us to improve our manuscript. We have addressed every comment from the reviewers and revised our manuscript accordingly. Please find our edits and responses to the reviewer comments below.

Reviewer 1: 

  1. Lines 56-149provide basic pregnancy physiology. Although the presented data are useful for general knowledge, they are not related to the topic of the review. This section should be either deleted or abbreviated.

Answer: Thank you for your feedback. We have significantly abbreviated this section. We feel that this section is very important for readers to understand how HEV is affecting normal pregnancy. In addition, because of the various loopholes in the HEV literature regarding pregnancy pathogenesis, we thought that this section would add up to the readers knowledge on pregnancy when compared with adverse effects listed in animal models. Thus, it would help to direct the future studies of HEV.

We have broken down section “1. Introduction to human pregnancy” in three subsections:

Line 41, 1.1 Three trimesters of pregnancy

Line 77, 1.2 Immunological changes during pregnancy to understand HEV related effects.

Line 113, 1.3 Hormonal changes during pregnancy to understand HEV related effects.

           We have abbreviated the sections as suggested by the reviewer.

  1. In addition, a summary of the reported cases in pregnancy should be provided with the target population, geographical location, detected genotype, the tested parameters, and the outcome of the infection at the level of mother and fetus/newborn should be presented.

Answer: We completely agree with the reviewer as they would want to see detailed epidemiological information on the pregnancy pathology by HEV. In this review, we have focused on the update of animal models from the very start to the end. Thus, listing every possible study that was done to find the perfect animal model.

We have another book chapter in press, releasing soon in April 2023 which has detailed epidemiological information on pregnancy cases, genotype, geographical location, and mortality cases.

  1. The tile, introduction to HEV, which includes lines 150-250 (about 100 lines) mentioned different topics related to HEV including the structure, genotypes, complications, and epidemiology in pregnant women. Therefore, this section should be divided into subtitles to make it easier for the readers to follow the presented data.

Answer: We would like to thank the reviewer for suggesting this idea. We have divided our sections as per the reviewer’s suggestions. We understand that our review is comparatively long as we wanted the reader to gain sufficient knowledge on the necessity of HEV animal models for studying pregnancy pathology.

We have broken down section “2. HEV background” in three subsections:

Line 163, 2.1 HEV new classification

Line 174, 2.2 Geographical distribution of HEV

Line 188, 2.3 Pregnancy mortality reported due to HEV

  1. I see that the authors have reported a review about the use of animal models in studying the replication and pathogenesis of HEV in general. Unfortunately, only limited data related to the topic of pregnancy are presented.Authors should focus on the main target of the manuscript rather than describing HEV in animal models which were extensively reviewed. I know that the data related to pregnancy are not much however, spotting the light on this information is of great value to readers. For example, the title 1.4. Rabbit Model (lines 287-477). Studies of HEV replication in the ovary and transmission in pregnancy are discussed only in lines 430-477!

Answer: We would like to thank the reviewer for going through our manuscript in detail. We would like to highlight that our manuscript is focused on emphasizing the animal trials that have been done to study the congenital transmission to demonstrate the pregnancy related pathology. Thus, even though it includes higher replication and pathogenesis in pregnancy related organs, we intend to describe the pregnancy pathological factors.

  1. Similar observations are seen in the other animal models.

Answer: We would like to thank the reviewer for bringing this to our attention.

  1. There are some important items that should also be included in this review such as studying the mechanisms responsible for the high morbidity of hepatitis E in pregnancy.
  • Focused on listing the update from the first animal study regarding pregnancy till date!

Answer: We would like to thank the reviewer for this wonderful suggestion of including mechanisms responsible for high morbidity of hepatitis E. We would like to bring into the attention of reviewer that we already have the “HEV immunopathogenesis” manuscript available on the link Hepatitis E Virus Immunopathogenesis - PubMed (nih.gov) which was published last year by us.

Because there is very minimum information detailing the primary cause of HEV induced pregnancy pathology, we intend to list every possible animal model study and its findings to provide the readers with the areas that are still left to be studied.

Reviewer 2: 

The manuscript "Animal models for studying congenital transmission of hepatitis E virus" describes HEV animal model systems and their relative pros and cons as regards the HEV pregnancy associated mortality. The manuscript is well written and detailed in its description.

Answer: We would like to thank the reviewer for the comments. We are glad that the reviewer enjoyed this manuscript, and we appreciate the motivational comments. 

Reviewer 3: 

Major Points

  1. The references and reports that are discussed in the ‘Introduction to HEV’ section appear to be somewhat dated with many of the highlighted studies being over 10 years old.  To make the manuscript as cutting-edge as possible, the inclusion and discussion of more recent reports is highly encouraged.

Answer: We would like to thank the reviewer for studying our manuscript in such detail. We agree with the reviewer that inclusion of recent papers would be valuable, but the cited papers are the initial manuscripts that defined structural facts, genotypes. Thus, the papers referenced are the primary sources such as that discovered the structural morphology of HEV or summarized the genotypes in the large geographical regions.

  1. Rabbit model section:  the references appear to focus on older studies and I believe stop at 2019.  There have been several interesting studies that have been reported over the last three years that should be included in the text. 

     Answer: Thank you for mentioning this part as a comment. This is one of the reasons we structured our manuscript in a pattern from when the initial study was done and what were the relevant findings that are important to understand the congenital transmission of HEV. This would help the reader to highlight the loopholes and understand the area where the new research can be directed.

     We have added one study regarding HEV pregnancy in rabbit model that we found was missing in the manuscript, but it was also done in 2019. There are no reports to our knowledge that was done in 2020, 2021 and 2022 regarding pregnancy pathology in rabbits.

Line 453 – 465 has been added to the manuscript.

     “In 2019, Li et al. studied the pregnancy results that could be pertaining to different HEV genotypes and if the prevention can be achieved by HEV239 (Hecolin, 26 kDa recombinant polypeptide expressed by the Escherichia coli system derived from the 368-606 amino acid segment of the HEV gt1 ORF2) vaccine. Forty-two female rabbits were divided in to 7 groups, with 2 groups were administered with preventative vaccine, 1 group with PBS. Except for the negative control, all other groups were either inoculated with rabbit HEV gt3 (CHN-BJ-R14) or swine HEV gt4 (CHN-SD-SW2) or human HEV gt3 (CHN-SH-W). Interestingly, pre-exposure to HEV 239 vaccine before copulation resulted in no HEV infections. However, all other rabbits inoculated with different strains from rabbit, swine and humans successfully resulted in adverse pregnancy outcomes. Furthermore, newborn rabbits born to vaccinated individuals were free of HEV in comparison to the survived individuals from HEV infected females demonstrated HEV infection”.  

  1. A minor point (that also applies to other animal model sections) – it would be optimal if instead of discussing individual studies one at a time, the authors would integrate the findings together into comprehensive paragraphs organized around key conclusions.  This alternative format would allow for a more efficient and effective reading experience for the general readership.

Answer: We would like to thank the reviewer for the wonderful suggestion. As per our experience, we find that is a normal trend of summarizing. However, in our opinion, we realized that it’s hard for the readers to track down the findings in series in a particular animal model where the reader must refer to multiple papers even after reading a review manuscript. Thus, as stated earlier, our intention was to structure our manuscript in a yearly pattern highlighting the initial study to the recent study and their relevant findings that are important to understand the congenital transmission of HEV. This would help the reader to highlight the loopholes and understand the area where the future research can be directed. Furthermore, for ease, we have summarized the major clinical manifestations in figures for readers to understand the limitation of different animal models in comparison to HEV in humans.

  1. Mouse model section:  the references are also somewhat outdated and are generally missing studies that have been reported over the last three years.

Answer: Thank you to the reviewer for mentioning this. We have included one study that we missed to include in our manuscript regarding adverse pregnancy outcomes in mouse model. We have now included the study in the manuscript.

Line 628 – 633 has been added to the manuscript.

“Furthermore, in 2021, Yang et al demonstrated miscarriage, non-pregnancy, and uterine damage in the HEV gt4 (KM01) infected BALB/c mice after copulation with the healthy males. They suggested uterine damage in relation to endometrial thinning, enhanced inflammatory response and increased programmed cell death would be causing factors of pregnancy pathology. Moreover, they demonstrated recovery of uterine layers and improvement in the fertility after the clearance of HEV from female BLAB/c mice”.

  1. Pig model section:  the references provided and discussed are also somewhat dated – I don’t believe that any papers from the last decade are discussed.

Answer: We would like to thank the reviewer for catching this major point that we as the writer want to make. As you have found that pig studies regarding congenital transmission or pregnancy pathology are from decades ago, because pregnancy mortality cannot be recapitulated in pigs. Thus, after the discovery of the first animal strain of HEV in swine in 1997, efforts were made to understand if HEV induces classical pregnancy pathology in pigs. However, only subclinical infection of HEV was noted in pigs. Pigs didn’t demonstrate any pregnancy mortality, fetal pathology or vertical transmission. Thus, in a race of being best animal model to understand the factors contributing to pregnancy mortality mimicking clinical HEV disease in humans, pigs were left behind. However, because pigs are the major known zoonotic reservoir for HEV, an experimental study regarding pregnancy and HEV replication has been described.

  1. Chicken section:  the same concern regarding dated references and a lack of work reported over the last decade/decade and a half applies.

Answer: We would like to thank the reviewer for the comment. Chicken has been mentioned as the only animal model that demonstrates liver pathology somewhat like humans during HEV induced pregnancy pathology. In addition, chicken also demonstrates the vertical transmission of HEV. There are no recent studies enhancing the literature on these subjects. Thus, the paper mentioned are the primary sources of the above-mentioned findings.

  1. In vitro models.  Only one paper is cited/discussed.  There has been additional work in this area of the field (e.g. PMID: 30420629) that I encourage the authors to include/discuss in the review.

     Answer: Thank you for bringing this out! We would be happy to include this paper. However, while addressing the reviewer 1 comments, we realized that we need to include only the animal models that describe the pregnancy related pathology highlighting the replication in the reproductive organs and evidence of vertical transmissions. Thus, we have removed the one and only included in vitro section (placenta model) of HEV. However, in our upcoming research paper regarding HEV induced liver pathology model in chickens, we will be including this paper while highlighting the human HEV infection.

Minor Points:

  1. Line 83 – there is an extra space between in and diminished

Answer: Took care of it!

  1. Line 152:  The discovery that HEV is ‘quasi- enveloped’ is an interesting one – perhaps the authors could consider adding a bit more of an explanation regarding the envelope of these historically thought to be non-enveloped virions for the general reader.

Answer: Line 145-146, “The absence of peplomers at the membrane associated HEV virions was the confirmation that the virus is quasienveloped, just like hepatitis A virus” has been added. It has been referenced for the readers who would like to go into much detail.

  1. Line 172:  There is a text size/font change that needs to be addressed (balayani)

Answer:Paslahepevirus balayani” has been italicized as it refers to the scientific name

  1. Line 194:  I’ve generally seen this presented as ‘fecal-oral’ rather than feco-oral.

Answer: Thank you for mentioning this. We have corrected it and used “fecal-oral” consistently in the manuscript.

  1. Line 195: data is a plural word (thus this should be written as data are rather than data is

Answer: The change has been made “data are…”

  1. Line 208:  reword the sentence at the start of the paragraph for clarity.

Answer: Line 205 – 208

Thank you for mentioning this, a line has been added to make it meaningful.It has been over three decades, since HEV was recognized as a reason of infectious hepatitis, several studies after that have similarly reported high rates of maternal, fetal, and neonatal illness and death in pregnancies affected by HEV, however, the mechanism for pregnancy mortality has not been identified yet.

  1. Line 210-213:  This sentence discuss a report from a ‘recent’ disease model – but all of the references are over a decade old.  Please reword and put this in the proper context.

Answer: Line 208

Thank you for catching this word. As per the suggestion, we have removed the word “recent”. We would like to mention to the reviewer that there is very minimal data regarding stillbirths and fetal deaths. Thus, the references mentioned are the most updated data.

  1. Fig. 1 and elsewhere:  The black lettering in a dark brown box is not optimal/easy to read.

- brown box “clinical manifestations” change!

Answer: We would like to thank the reviewer for pointing this out. We have made the changes to a light color!. Changes have been made in all the figures.

  1. Section 1.3 – use a different system (maybe brackets?) for the numbers that reflect criteria for an optimal model.  They are currently presented in the same format in the text as are references.

- criteria change

Answer: Line 254-285

We would like to thank the reviewer for indicating this! We have made the changes, (a), (b), (c)….

  1. Section 1.3:  wouldn’t an additional characteristic of an optimal HEV model be highly pathology in the pregnant animal versus non-pregnant test subjects?

Answer: Getting this question from the reviewer makes us happy as our review is well understood by the reviewer. The same point has been made in the second characteristic written as “classical pathological symptoms”. However, we have included “pregnancy pathology”, so the readers would get the idea.

Line 260,

“The model should recapitulate HEV pathology including liver lesions and similar sites of replication. To fully recapitulate the human condition, researchers should strive to reproduce the classical pathological symptoms (pregnancy pathology) associated with the disease.  Simply injecting a virus into the placenta of a pregnant animal has a high likelihood of causing abortion and other morbidities but likely does not mimic a true systemic infection”.

  1. Line 507 – there is an extra space between in and humans

Answer: This has been taken care of.

Reviewer 2 Report

The manuscript "Animal models for studying congenital transmission of hepatitis E virus" describes HEV animal model systems and the their relative pros and cons as regards the HEV pregnancy associated mortality. The manuscript is well written and detailed in its description.  

Author Response

(The authors gave the same response as above.)

Reviewer 3 Report

                This is an overview of pregnancy and HEV infection with an emphasis on animal model systems.  Overall I find the manuscript to be interesting and informative.  However as noted below, I do find aspects of the manuscript to focus on older literature and not to include work published in the area over the last couple of years.  This lack of treatment/discussion of up to date studies could limit the impact of the manuscript. 

Major Points

1.        The references and reports that are discussed in the ‘Introduction to HEV’ section appear to be somewhat dated with many of the highlighted studies being over 10 years old.  To make the manuscript as cutting-edge as possible, the inclusion and discussion of more recent reports is highly encouraged.

2.       Rabbit model section:  the references appear to focus on older studies and I believe stop at 2019.  There have been several interesting studies that have been reported over the last three years that should be included in the text. 

3.       A minor point (that also applies to other animal model sections) – it would be optimal if instead of discussing individual studies one at a time, the authors would integrate the findings together into comprehensive paragraphs organized around key conclusions.  This alternative format would allow for a more efficient and effective reading experience for the general readership.

4.       Mouse model section:  the references are also somewhat outdated and are generally missing studies that have been reported over the last three years.

5.       Pig model section:  the references provided and discussed are also somewhat dated – I don’t believe that any papers from the last decade are discussed.

6.       Chicken section:  the same concern regarding dated references and a lack of work reported over the last decade/decade and a half applies.

7.       In vitro models.  Only one paper is cited/discussed.  There has been additional work in this area of the field (e.g. PMID: 30420629) that I encourage the authors to include/discuss in the review

Minor Points:

1.        Line 83 – there is an extra space between in and diminished

2.       Line 152:  The discovery that HEV is ‘quasi- enveloped’ is an interesting one – perhaps the authors could consider adding a bit more of an explanation regarding the envelope of these historically thought to be non-enveloped virions for the general reader.

3.       Line 172:  There is a text size/font change that needs to be addressed (balayani)

4.       Line 194:  I’ve generally seen this presented as ‘fecal-oral’ rather than feco-oral.

5.       Line 195:  data is a plural word (thus this should be written as data are rather than data is

6.       Line 208:  reword the sentence at the start of the paragraph for clarity.

7.       Line 210-213:  This sentence discuss a report from a ‘recent’ disease model – but all of the references are over a decade old.  Please reword and put this in the proper context.

8.       Fig. 1 and elsewhere:  The black lettering in a dark brown box is not optimal/easy to read.

9.       Section 1.3 – use a different system (maybe brackets?) for the numbers that reflect criteria for an optimal model.  They are currently presented in the same format in the text as are references.

10.   Section 1.3:  wouldn’t an additional characteristic of an optimal HEV model be highly pathology in the pregnant animal versus non-pregnant test subjects?

11.   Line 507 – there is an extra space between in and humans

Author Response

(The authors gave the same response as above.)

Round 2

Reviewer 3 Report

The authors have adequately addressed the points raised in the initial round of critiques.  While I would still like to see the inclusion of more recent references to augment the primary original references cited and fully present the current status of the field, I will defer to the authors on this point.

Author Response

Thank you